# Mitochondrial Dysfunction: A Common Hallmark Underlying Comorbidity between sIBM and Other Degenerative and Age-Related Diseases

**DOI:** 10.3390/jcm9051446

**Published:** 2020-05-13

**Authors:** Marc Catalán-García, Francesc Josep García-García, Pedro J. Moreno-Lozano, Gema Alcarraz-Vizán, Adrià Tort-Merino, José César Milisenda, Judith Cantó-Santos, Tamara Barcos-Rodríguez, Francesc Cardellach, Albert Lladó, Anna Novials, Glòria Garrabou, Josep M. Grau-Junyent

**Affiliations:** 1Muscle Research and Mitochondrial Function Laboratory, CELLEX-IDIBAPS, Faculty of Medicine, University of Barcelona, 08036 Barcelona, Spain; marccatalangarcia@gmail.com (M.C.-G.); fjgarcia@ub.edu (F.J.G.-G.); pjmoreno@clinic.cat (P.J.M.-L.); jcmilise@clinic.cat (J.C.M.); jcanto@clinic.cat (J.C.-S.); barcos@clinic.cat (T.B.-R.); fcardell@clinic.cat (F.C.); jmgrau@clinic.cat (J.M.G.-J.); 2Internal Medicine Department, Hospital Clinic of Barcelona, 08036 Barcelona, Spain; 3CIBERER—Spanish Biomedical Research Centre in Rare Diseases, 28029 Madrid, Spain; 4Diabetes and Obesity Laboratory Research, Institut d’Investigacions Biomèdiques August Pi I Sunyer (IDIBAPS), 08036 Barcelona, Spain; alcarraz@clinic.cat; 5CIBERDEM—Spanish Biomedical Research Centre in Diabetes and Associated Metabolic Disorders, 28029 Madrid, Spain; 6Alzheimer’s Disease and Other Cognitive Disorders Unit, Hospital Clínic and Institut d’Investigacions Biomèdiques August Pi I Sunyer (IDIBAPS), 08036 Barcelona, Spain; atort@clinic.cat

**Keywords:** sIBM 1, Alzheimer 2, T2DM 3, mitochondria 4, comorbidity 5, myositis 6

## Abstract

Sporadic inclusion body myositis (sIBM) is an inflammatory myopathy associated, among others, with mitochondrial dysfunction. Similar molecular features are found in Alzheimer’s disease (AD) and Type 2 Diabetes Mellitus (T2DM), underlying potential comorbidity. This study aims to evaluate common clinical and molecular hallmarks among sIBM, AD, and T2DM. Comorbidity with AD was assessed in *n* = 14 sIBM patients by performing neuropsychological and cognitive tests, cranial magnetic resonance imaging, AD cerebrospinal fluid biomarkers (levels of amyloid beta, total tau, and phosphorylated tau at threonine-181), and genetic apolipoprotein E genotyping. In the same sIBM cohort, comorbidity with T2DM was assessed by collecting anthropometric measures and performing an oral glucose tolerance test and insulin determinations. Results were compared to the standard population and other myositis (*n* = 7 dermatomyositis and *n* = 7 polymyositis). Mitochondrial contribution into disease was tested by measurement of oxidative/anaerobic and oxidant/antioxidant balances, respiration fluxes, and enzymatic activities in sIBM fibroblasts subjected to different glucose levels. Comorbidity of sIBM with AD was not detected. Clinically, sIBM patients showed signs of misbalanced glucose homeostasis, similar to other myositis. Such misbalance was further confirmed at the molecular level by the metabolic inability of sIBM fibroblasts to adapt to different glucose conditions. Under the standard condition, sIBM fibroblasts showed decreased respiration (0.71 ± 0.08 vs. 1.06 ± 0.04 nmols O_2_/min; *p* = 0.024) and increased anaerobic metabolism (5.76 ± 0.52 vs. 3.79 ± 0.35 mM lactate; *p* = 0.052). Moreover, when glucose conditions were changed, sIBM fibroblasts presented decreased fold change in mitochondrial enzymatic activities (−12.13 ± 21.86 vs. 199.22 ± 62.52 cytochrome c oxidase/citrate synthase ratio; *p* = 0.017) and increased oxidative stress per mitochondrial activity (203.76 ± 82.77 vs. −69.55 ± 21.00; *p* = 0.047), underlying scarce metabolic plasticity. These findings do not demonstrate higher prevalence of AD in sIBM patients, but evidences of prediabetogenic conditions were found. Glucose deregulation in myositis suggests the contribution of lifestyle conditions, such as restricted mobility. Additionally, molecular evidences from sIBM fibroblasts confirm that mitochondrial dysfunction may play a role. Monitoring T2DM development and mitochondrial contribution to disease in myositis patients could set a path for novel therapeutic options.

## 1. Introduction

Inflammatory myopathies are a heterogenic group of diseases characterized by a progressive symmetric weakness and inflammatory infiltrates within the skeletal muscle [1,2]. Regarding their distinct clinicopathological features, inflammatory myopathies can be classified in four main subtypes: dermatomyositis (DM), polymyositis (PM), necrotizing autoimmune myopathies (NAM), and sporadic inclusion body myositis (sIBM) [3].

The former, sIBM, is the unique inflammatory myopathy associated with the accumulation of misfolded proteins and ageing [2]. With a male:female ratio of 3:1, it is considered a rare disease (ORPHA611) and its prevalence varies from 4.7 per million in the Netherlands [4] to 50 per million in South Australia [5]. In fact, these rates continue to increase, probably due to improved diagnostic protocols and increased ageing of the population, making it the most frequent myopathy in individuals over their 50s, at least in some countries.

In stark contrast with other myositis, sIBM patients show asymmetrical wasting of proximal and distal muscles, in particular quadriceps and finger flexors, leading to difficulty in performing tasks requiring these muscles such as lifting objects or climbing stairs, among others. In addition, neck extensors and pharyngeal involvement may cause dropped head or dysphagia, respectively, affecting 60% of patients over time [2,5].

At the moment, muscle biopsy remains the definitive diagnostic procedure for sIBM [6]. At the histopathological level, the combination of inflammatory changes (CD8+ T-cell infiltrate and expression of class I major histocompatibility complex antigens -MHC-I- in muscle cells), degenerative features (rimmed vacuoles and intra-cellular protein aggregates) [7], and mitochondrial changes (COX-/SDH+ cells and ragged-red-fibbers) are specific and required for sIBM diagnosis [8]. Although knowledge about the molecular hallmarks of the disease is increasing every day, sIBM is still a disease without an effective treatment. Moreover, the aetiology of this progressive degenerative disorder of the skeletal muscles remains unknown [2].

Interestingly, the presence of inflammation, protein depots, and mitochondrial failure seems not to be exclusive for sIBM and, in the last few years, growing evidence has supported the deregulation of these pathways in other chronic and age-related neurodegenerative or metabolic disorders. Among them, the deregulation of mitochondrial function and associated cell consequences has been speculated as the major molecular hallmark underlying a number of these disorders, since mitochondrial dysfunction can trigger, by its own, from bioenergetic failure and oxidative stress production to protein accumulation, inflammation, and cell death [9], features that are present in most of these degenerative age-related disorders.

In physiologic conditions, mitochondria are essential for cell survival, playing a central role in calcium homeostasis, bioenergetics, metabolism, and cell respiration. In pathological conditions, they are indeed the main organelle involved in triggering apoptosis and producing reactive oxygen species (ROS).

Thus, mitochondrial homeostasis and balance is fundamental for cell function, especially in highly energetic and dependent tissues (including muscle, central nervous system, lymphoid organs, or glands), where its dysfunction may become evident much earlier.

Despite the fact that pathological mitochondrial evidence is frequently observed in muscle biopsies from sIBM patients, mitochondrial abnormalities in this disease have scarcely been assessed at a molecular level [10,11]. Our group described, for the first time, mitochondrial DNA alterations in sIBM muscle. Furthermore, down-regulated activity of mitochondrial respiratory chain (MRC) has been found, not only restricted to muscle, also in other peripheral tissues, such as blood [12].

Interestingly, mitochondrial abnormalities, energy failure, respiratory chain impairment, and generation of ROS have also been described in brain aging related to neural loss and synaptic alteration [13]. Moreover, the accumulation of a variety of proteins such as β-amyloid, phospho-Tau, p62, TDP43, and caveolin, among others, and consequent inflammation in the muscle of sIBM patients is like those observed in the brains of patients with neurodegenerative disorders, such as Alzheimer’s disease (AD). For all these common molecular hallmarks, growing support is given to a potential shared pathophysiology between sIBM and AD [14,15]. In fact, sIBM is also known as the “muscle Alzheimer’s disease” due to these common deregulations. However, as far as we know, neither epidemiological nor molecular studies have been conducted so far to corroborate or discard the comorbidity between these two disorders.

Alzheimer’s disease is the commonest cause of dementia, with huge implications for individuals and large sociosanitary and economic associated burdens currently affecting 50 million people. AD is the single biggest cause of dementia, accounting for 50%–75%, and is primarily a condition of later life, roughly doubling in prevalence every 5 years after age 65 [16]. Dementia is the major cause of dependence, disability, and mortality. In the coming years, the largest increase in dementia prevalence is expected to affect 152 million by 2050, which indeed shows patterns of association with cardiovascular disease, hypertension, and diabetes [17]. Consequently, any data that may support comorbidity with alternative diseases may be of help to guide assistance and improve the quality of life in these patients.

Similarly to sIBM and AD, the accumulation of proteins and inflammation has also been described as a trigger of type 2 diabetes mellitus (T2DM). In fact, the presence of amyloid deposits within the pancreatic islets is a pathophysiological hallmark of T2DM [18]. Moreover, mitochondrial dysfunction concerning bioenergetics alterations, respiratory chain disability, and ROS production in T2DM has also been widely described in different tissues [19,20].

T2DM is one of the most prevalent and chronic metabolic disorders worldwide. It is characterized by the presence of hyperglycaemia caused by different alterations converging in the development of insulin resistance accompanied by relative to severe insulin deficiency secretion [21]. This disease is associated with being overweight in 90% of cases, a family history of T2DM in 75%–90% of cases, and poor insulin production. It was estimated that in 2017 there were 451 million people with diabetes worldwide. These figures were expected to increase to 693 million by 2045. Currently, approximately 5 million deaths worldwide were attributable to diabetes [22] and its frequency is expected to increase on account of extensive diagnosis and aging of the population in developed countries.

The link between AD and T2DM is clearly established in literature [23,24]. However, despite the fact that there are common molecular mechanisms that are also shared with sIBM, the comorbidity between T2DM and sIBM has not yet been clarified.

In this study, we aim to evaluate sIBM patients at two levels to assess clinical and molecular comorbidity between sIBM, AD, and T2DM. From a clinical point of view, we will evaluate the neurological and metabolic status in a cohort of sIBM patients to determine the comorbidity between sIBM and AD or T2DM, respectively. In addition, we aim to include a cohort of patients with inflammatory myopathies other than sIBM, such as DM and PM, to evaluate differential or common deregulation with diseases that clinically overlap with sIBM. From a molecular point of view, we aim to evaluate common molecular hallmarks between these degenerative and age-related diseases, focusing our interest on the mitochondrial homeostasis of sIBM fibroblasts that will be subjected to different pathophysiological conditions.

## 2. Experimental Section

### 2.1. Study Population

The sIBM cases were diagnosed by clinical and pathological tests in the Department of Internal Medicine from the Hospital Clínic of Barcelona (Barcelona, Spain). All the patients fulfilled the criteria proposed by the European Neuromuscular Centre for sIBM diagnosis [25]. Exclusion criteria were: age < 40 years, family history of hereditary mitochondrial disease and HIV infection, or drug abuse. A cohort of 14 patients was consequently included after signing the informed consent previously approved by the Ethical Committee of our institution (code HCB/2015/0562). Then, 10 underwent neurological evaluation and 8 (not necessarily the same participants) underwent T2DM and mitochondrial testing, depending on their possibilities to endure invasive or uncomfortable tests such as magnetic resonance imaging (MRI), lumbar puncture, or biopsy performance.

On inclusion, sIBM patients completed the IBMFRS (inclusion body myositis functional rating scale), a validated disease-specific functional rating scale to assess disease severity. Epidemiological and clinical data (age, gender, and ethnicity) were also recruited.

#### 2.1.1. Biopsy Performance

Muscles biopsies were performed for diagnostic purposes indicated for muscle weakness or raised creatine kinase (CK) levels. Surgical muscle biopsies were obtained by trained physicians and the samples were processed routinely in the laboratory as described elsewhere [26].

#### 2.1.2. Histological Studies

Fresh samples were frozen in cooled isopentane, sectioned by cryotome at −30 °C, and stained and reacted with different reagents for diagnostic purposes: haematoxylin and eosin, modified Gomori’s trichrome, non-specific esterase, periodic acid-Schiff stain, Oil Red O, acid and alkaline phosphatase, NADH, COX, SDH, and ATPase at pH 4.3, 4.6 and 9.4, as reported elsewhere [26]. In addition, some immunohistochemistry reactions such as MHC-I as well as p62 were performed. In parallel, blood samples or cerebrospinal fluid (CSF) from sIBM patients were collected, when indicated, for analytic purposes.

#### 2.1.3. Control Population

According to the standards for diagnosis, results of clinical evaluation for T2DM and AD in sIBM patients were compared to age bracket control data from individuals free of disease or, alternatively, with a cohort of patients with inflammatory myopathies other than sIBM, such as DM (*n* = 7) and PM (*n* = 7). This second cohort was prospectively included in parallel to sIBM patients to evaluate differential or common deregulation with diseases that clinically overlap with sIBM. These patients were also diagnosed at the Department of Internal Medicine from the Hospital Clinic of Barcelona (Barcelona, Spain) following the current established clinical and pathological criteria [3]. Experimental studies for mitochondrial evaluation were performed in the same cohort of sIBM patients and compared to healthy control volunteers (C; *n* = 5), who were prospectively included.

### 2.2. Neurological Evaluation of sIBM Patients

This part of the study included neuropsychological and cognitive tests, cranial MRI, AD biomarkers in CSF (amyloid beta or Aß42 levels, total tau, and phosphorylated tau at threonine-181 content), and apolipoprotein E (APOE) genotype.

#### 2.2.1. Clinical, Psychological, and Cognitive Assessment

Participants were evaluated by a trained neurologist with a comprehensive neuropsychological battery of tests. Clinical diagnosis was established following the current clinical criteria for mild cognitive impairment (MCI) [27] and dementia due to AD [28].

The neuropsychological battery of tests encompassed five cognitive domains. The memory domain included the Free and Cued Selective Reminding Test [29], the landscape test [30], and the Rey–Osterrieth complex figure [31]. The language domain comprised of the Boston Naming Test [32] and a semantic fluency task [33]. The praxis domain contained a copy of the Rey–Osterrieth complex figure and the ideomotor subtest of the Western Aphasia Battery [34]. The visual perception domain included the object decision, number location, and incomplete letters subtests of the Visual Object and Space Perception battery [35]. The executive functions domain consisted of the Trail Making Test [36], the Stroop Test [37], the Symbol Digit Modalities Test [38], and the Digit Span test of the Wechsler Adult Intelligence Scale [39]. Global cognition was assessed with the Mini Mental State Evaluation (MMSE) [40]. Premorbid intelligence was assessed with the Spanish word accentuation test [41]. Cognitive reserve was evaluated with the cognitive reserve questionnaire [42]. Finally, anxiety and depression levels were measured with the Hospital Anxiety and Depression Scale.

#### 2.2.2. Image Acquisition

For each participant, a T1-weighted, high-resolution, MPRAGE structural MRI scan (echo time (TE) 2.98 ms, repetition time (TR) 2300 ms, inversion time 900 ms, flip angle 9°, bandwidth 240 Hz/pixel, matrix 256 × 256, 240 axial slices, isometric voxel size 1⁄4 1.0 mm^3^) was acquired at the IDIBAPS’s Imaging core facilities with a 3T whole-body MRI scanner (Siemens Magnetom Trio; Hospital Clinic, Barcelona, Spain).

For all subjects, global cortical atrophy (GCA), medial temporal lobe atrophy (MTL), and white matter integrity (Fazekas) scores were obtained by a trained radiologist.

#### 2.2.3. Determination of AD CSF Biomarkers and Apolipoprotein E (APOE) Genotype

Subjects underwent a lumbar puncture between 9 a.m. and 12 p.m. In the extraction, 10 mL of CSF was collected. The samples were centrifuged and stored in polypropylene tubes at –80 °C within the first hour after extraction. CSF Aß42 levels, total tau (tau), and phosphorylated tau at threonine-181 (ptau) were measured by enzyme-linked immunosorbent assay kits (Innogenetics, Ghent, Belgium). The following cut off points were considered: (a) Aß42 ≤ 550 pg/mL, (b) tau ≥ 385 pg/mL, and (c) ptau ≥ 65 pg/mL.

Genomic DNA was extracted from the peripheral blood of probands using the QIAamp DNAblood minikit (Qiagen AG, Basel, Switzerland). APOE genotyping was performed by polymerase chain reaction amplification and HhaI restriction enzyme digestion.

### 2.3. T2DM Evaluation of sIBM Patients

#### 2.3.1. Oral Glucose Tolerance Test (OGTT) Procedures

Procedures for the 2h-OGTT required previous fasting after midnight, to obtain baseline fasting glucose level, and further administration of the oral glucose load (75 g) within a 5 min period. Blood specimens to determine the plasma glucose level were subsequently drawn at 30, 60, 90, and 120 min, timed from the beginning of the glucose load. Patients were established as having impaired fasting glucose (IFG; if the venous plasma glucose level was greater than 100 and less than 126 mg/dL) or having impaired glucose tolerance (IGT; if the 120-min venous plasma glucose value fell between 140 and 200 mg/dL). Criteria for new-onset T2DM was a fasting plasma glucose level greater than 126 mg/dL or 120-min venous plasma glucose level greater than 200 mg/dL on the OGTT.

#### 2.3.2. Medical Records Data

The following data were collected from medical records: circulating baseline levels of insulin (mU/L) and glycated haemoglobin, HbA1c (%). Homeostatic model assessment of insulin resistance (HOMA-IR) was calculated to determine the insulin resistance as: (basal insulin level in µU/mL × basal glucose level in mmol/L)/22.5. Anthropometric and physical examination data concerning weight and height were also recruited to calculate body mass index (BMI; calculated as weight in kilograms divided by the square of the height in meters).

### 2.4. Mitochondrial Characterization in sIBM Patients

#### 2.4.1. Fibroblasts Culture

Fibroblasts were obtained by a skin punch biopsy. Cells were grown in 25 mM glucose Dulbecco’s Modified Eagle’s medium (DMEM from Gibco, Life Technologies, Burlington, ON, Canada) supplemented with 10% heat-inactivated foetal bovine serum and 1% penicillin-streptomycin at 37 °C in a humidified 5% CO_2_ air incubator until 80% optimal confluence was reached. Fibroblasts were harvested with 2.5% trypsin (Gibco, Life Technologies, Burlington, ON, Canada) and centrifuged at 500 g for 8 min. All functional assays were performed in cells between passages 5 and 10.

To test fibroblast response to pro-diabetogenic in vitro conditions, cells were cultured under different glucose conditions: high glucose (HG: 25 mM) and low glucose (LG: 5 mM) DMEM media 10 days prior the experiment, mimicking the glucose deregulation observed in T2DM.

Expected response to 10 days glucose decrease (from HG to LG) should promote mitochondrial activation to enhance energetic output. To evaluate mitochondria activation in front of glucose reduction, we tested oxidative vs. anaerobic metabolism and oxidant/antioxidant balance, together with mitochondrial respiration and enzymatic activities.

#### 2.4.2. Mitochondrial Respiration

Mitochondrial respiration is the result of oxygen consumption by the MRC. It was measured in 1 million cells by high-resolution respirometry using Oroboros™ Oxygraph-2K^®^ technology (Innsbruck, Austria) in non-permeabilized fibroblasts, following the manufacturer’s protocols [43]. Two measures were obtained: one of them is considered a sign of proper mitochondrial function and is calculated as the ratio between the basal respiration flux (standard oxygen consumption) in respect to the maximal respiration capacity (by adding the uncoupler CCCP). In parallel, proton leak was secondarily measured (by adding oligomycin) as a sign of inefficient and unhealthy MRC activity. Results were expressed relative to cell count and were expressed as nmols/min.

#### 2.4.3. Lactate Production

Lactate measures were performed in 100 µL of fresh supernatant fibroblast culture using the epoc^®^ Blood Analysis System, epoc System Software version 3.32.0 (Siemens Healthineers Global, Erlangen, Germany), normalized by cell count and expressed as mmols/L.

#### 2.4.4. Total Antioxidant Capacity

Total Antioxidant Capacity (TAC) was measured in 100 µL of cell supernatant using an OxiSelect™ Total Antioxidant Capacity Assay kit (Cell Biolabs Inc., San Diego, CA, USA) by spectrophotometry (absorption maximum at 490 nm), normalized by cell count and expressed as µM CRE (Copper Reducing Equivalents).

#### 2.4.5. MRC Enzymatic Activity

Cytochrome c oxidase (COX) function was assessed in 40 µg of total protein from each fibroblast sample according to standardized protocols [44] and normalized by mitochondrial content, measured as citrate synthase (CS) enzymatic activity. CS is a mitochondrial enzyme from the Krebs cycle and is widely considered to be a reliable marker of mitochondrial amount. Both enzymatic activities were measured by spectrophotometry, using internal controls, and were expressed as the ratio between COX vs. CS function.

#### 2.4.6. Oxidative Stress Assay

Lipid peroxidation (an indicator of oxidative damage produced by ROS to cellular lipid compounds) was quantified in 150 µg of protein from a fibroblast sample using the OxysResearch^™^ LPO-586^™^ kit (Deltaclon, Portland, OR, USA) through the spectrophotometric measurement of malondialdehyde and 4-hidroxyalkenal, both of which are products of fatty acid peroxide decomposition. Lipid peroxidation was normalized by COX/CS activity as a measure of oxidative stress production per mitochondrial activity.

### 2.5. Statistics

Results were expressed as mean ± standard error of the mean (SEM) or in fold change between conditions (HG vs. LG). Statistical analysis was performed after filtering for outliers to evaluate differential phenotypes depending on group-assignment (patients or controls) by using the non-parametric independent sample Mann-Whitney U test. The signification threshold was set at *p* < 0.05.

## 3. Results

### 3.1. Clinical DATA

The sIBM patients included in this study corresponded to *n* = 14 subjects aged between 48 and 90 (mean age 67.70 ± 3.84 years), 9:5 male to female ratio, and all white Caucasian. According to the design of the study, no statistical differences were observed in regards to age, gender, or ethnicity between sIBM patients and the control cohorts. Specifically, PM patients presented a mean age of 69.57 ± 5.39 years and a 1:6 male to female ratio, DM subjects a mean age of 66.67 ± 3.20 years and a 2:5 male to female ratio, and healthy control volunteers aged 57.12 ± 5.38 years and presented a 1:4 male to female ratio, all white Caucasian, according to sIBM demographic characteristics.

At the time of inclusion, the sIBM patient cohort scored 22.00 ± 2.37 out of 40 with the IBMFRS test, indicating moderate-to-advanced sIBM clinical severity.

### 3.2. Neurological Evaluation of sIBM Patients

#### 3.2.1. Patient Characteristics

Ten sIBM patients underwent the neurological assessment. Education level ranged between 7 and 18 years and the cognitive reserve questionnaire scored 7 to 18 points. Demographics of the cognitive study are shown in Table 1.

#### 3.2.2. Clinical and Neuropsychological Assessment

Eight out of the ten subjects that underwent clinical and neuropsychological evaluation were cognitively impaired. From those, 5 were diagnosed with MCI and 3 with dementia (two of them were diagnosed as AD).

Three subjects presented symptoms of anxiety and/or depression. MMSE ranged between 6 and 29 (maximum MMSE score in health is 30 points) and the neuropsychological profile of the study subjects was heterogeneous, although most of them showed an amnestic pattern (verbal and/or visual memory decline).

Despite reported disturbances in the clinical and neuropsychological assessment, no higher incidence of dementia or AD was detected in sIBM patients when the results were compared to the prevalence of dementia in Europe [45]. See clinical and neuropsychological data of the Neurological study in Table 1.

#### 3.2.3. MRI Data

Eight sIBM subjects underwent the magnetic resonance scan. Medial temporal lobe atrophy (MTL score > 1) was found in 3 subjects (maximum pathological score is 4 points) and all individuals presented different levels of white matter deterioration, including two subjects with a Fazekas score ≥ 2 (maximum pathological score is 3) (see Table 1).

However, MTL score > 1 was also found in controls in previous studies [46]. Furthermore, in our study, 1 out of 2 subjects with MTL >1 had normal AD CSF biomarkers, thus excluding AD [27]. Consequently, we concluded that MRI analysis of sIBM patients reported no higher incidence of AD.

#### 3.2.4. AD CSF Biomarkers and APOE Genotype

Five subjects completed the CSF testing for AD biomarkers and APOE genotype. CSF biomarker levels were positive for AD in one case and, additionally, one subject was an APOE ε4 carrier. From the 3 dementia cases, one presented a typical AD CSF profile, another received a clinical diagnosis of AD without biological confirmation, and the last one presented advanced dementia at 90 years. None of the MCI had a typical AD CSF biomarker pattern, although one subject presented reduced levels of CSF Aß_42_ with normal CSF tau and phosphor-tau levels (amyloidosis alone; data regarding AD CSF biomarkers and APOE genotype is shown in Table 1).

We found only 1 out of 5 patients with typical AD CSF biomarkers and this percentage is very similar to the frequencies of brain amyloidosis found in individuals with normal cognitive function at similar ages [47].

Considering the diagnostic criteria stablished by the National Institute on Aging-Alzheimer’s Association workgroups [27], neither CSF biomarkers nor APOE genotyping stand for higher prevalence of AD in sIBM patients when compared with age bracket control data.

### 3.3. T2DM Evaluation of sIBM Patients

The OGTT was performed to *n* = 8 sIBM patients with respect to the normality ranges of age bracket control data [48]. When basal glucose level was measured, 38% (3/8) of patients showed disturbances: 25% (2/8) were in the prediabetic range (plasma glucose level greater than 100 and less than 126 mg/dL), while the other 13% (1/8) were considered as T2DM patients (Figure 1).

The 2 h venous plasma glucose level in the sIBM cohort (*n* = 7) showed alterations in 42% (3/7) of the patients, while 14% (1/7) were in the prediabetic range (the plasma glucose level fell between 140 and 200 mg/dL) and the remaining 28% (2/7) felt into the T2DM category (glucose level greater than 200 mg/dL). Moreover, medical records in blood tests from the sIBM population (*n* = 8) showed increased HbA1c % in 50% (4/8) of these patients (HbA1c greater than 5.7). Finally, when HOMA-IR was assessed in the sIBM cohort (*n* = 7), 14% (1/7) of the patients were close to insulin resistance values (HOMA-IR > 5; Figure 1).

With respect to the age bracket control data [48], all evaluated sIBM patients presented one or more deregulated parameter of glucose metabolism, suggesting a prediabetogenic state. Moreover, when sIBM subjects were compared with respect to other patients presenting with alternative myopathies (DM and PM), similar deregulation of glucose parameters was observed, as shown in Figure 1 and Table 2.

Specifically, the basal glucose level was also determined in a cohort of patients with DM (*n* = 7) and PM (*n* = 7) showing prediabetic state values in 43% (3/7) of cases from both populations. The 2 h venous plasma glucose level in the DM population was greater in 43% (3/7) of patients, which is considered suggestive of a prediabetic state, while the PM cohort showed altered values in 71% (5/7) of patients, 28% (2/7) were considered as in a prediabetic state, and the other 43% (3/7) were consistent with T2DM. Medical data records from blood tests of DM and PM populations also showed altered HbA1c profiles in both cohorts. Particularly, 14% (1/7) of DM and 43% (3/7) of PM had greater values compared to age bracket control data (HbA1c < 5.7%) [48]. Finally, the HOMA-IR calculation showed that 14% (1/7) of the DM cohort and 43% (3/7) of the PM population showed greater HOMA-IR ratio than age bracket control ranges, however the obtained data is under the insulin resistance values (HOMA-IR > 5).

Considering the T2DM definition stratified by age from the American College of Endocrinology [49], the alteration of T2DM biomarkers in sIBM patients (and in the rest of the studied cohorts as PM and DM) is consistent with a prediabetic condition, eventually demonstrating the clinical comorbidity between these diseases.

### 3.4. Mitochondrial Characterization in sIBM Patients

Under standard conditions (see Figure 2), sIBM fibroblasts showed significantly decreased basal to maximal respiration (0.71 ± 0.08 vs. 1.06 ± 0.04 nmols/min; *p* = 0.024), increased anaerobic metabolism (5.76 ± 0.52 vs. 3.79 ± 0.35 mM lacate; *p* = 0.052), and TAC levels (403.89 ± 107.02 vs. 171.73 ± 23.17 µM CRE; *p* = 0.006), reflecting the mitochondrial dysfunction characteristic of the target tissue of the disease (muscle).

Cells increase mitochondrial activity in response to the decrease of glucose levels, probably as an attempt to maximize the energetic output in case of glucose restriction. Such activation was observed in control fibroblasts when subjected to glucose changing conditions (HG vs. LG) in terms of respiration. Under high glucose concentration, the mitochondrial respiration rate was low, indicating decreased oxygen consumption and high anaerobic metabolism. When the glucose concentration was diminished, oxygen consumption increased, indicating the activation of the aerobic (mitochondrial) metabolism (Appendix A).

Contrarily, when sIBM fibroblasts were subjected to glucose decrease to evaluate mitochondrial contribution to sIBM-T2DM comorbidity, such activation was not observed. Specifically, sIBM fibroblasts vs. controls showed decreased fold change basal to maximal respiration (1.65 ± 15.74 vs. 34.88 ± 58.02) and enzymatic activities (−12.13 ± 21.86 vs. 199.22 ± 62.52 COX/CS ratio; *p* = 0.017). Additionally, a lack of mitochondrial activation in sIBM fibroblasts was accompanied by increased fold change proton leak (92.29 ± 73.89 vs. 6.16 ± 83.05) and oxidative stress per mitochondrial activity (203.76 ± 82.77 vs. −69.55 ± 21.00; *p* = 0.047) (Figure 3).

Glucose misbalance manifested at a clinical level in sIBM patients was further confirmed in vitro by the inability of sIBM fibroblasts to adapt to different glucose conditions, thus underlying the clinical and molecular comorbidity between sIBM and T2DM.

## 4. Discussion

Despite medical literature postulating potential comorbidity of sIBM with AD and T2DM, scarce data and epidemiological studies (if any) had been conducted to date to evaluate such overlap. Before our investigation, none of the sIBM patients had been diagnosed of T2DM or AD. In the present study, and regardless of the small sample size herein tested, we found null comorbidity between sIBM and AD, but trends towards comorbidity between sIBM and T2DM.

Despite sIBM having been described as “Alzheimer in muscle” and both disorders sharing some pathophysiological mechanisms [14,15,50,51], the present outputs suggest that AD pathology is not more frequent in sIBM patients than in the age-paired population. These findings suggest that despite both diseases sharing common molecular pathways (including mitochondrial dysfunction, protein deposition, and inflammation, among others), such disorders present different regulatory mechanisms and/or the disease is specific, in each case, to a different target tissue.

Interestingly, partial comorbidity between sIBM and T2DM has been demonstrated in this work. Clinically, the metabolic deregulation in sIBM patients is characterized by the initial impairment of glucose metabolism compared with age bracket control data. Despite the fact that only a few of them could be concomitantly diagnosed by sIBM and T2DM, most of the sIBM patients showed clear signs of a prediabetogenic condition that underlie the comorbidity between both diseases.

Such coexistence of T2DM in sIBM patients could be due to common etiological links between both disorders. Previous literature describes similar mitochondrial alterations in T2DM and sIBM [5,20,52,53].

At a molecular level, the mitochondrial characterization of fibroblasts from sIBM patients under standard conditions of the present study demonstrate increased anaerobic metabolism and antioxidant capacity at the detriment of mitochondrial oxygen consumption. These results suggest that mitochondrial dysfunction is present in sIBM patients and is not only restricted to the target tissue of the disease (muscle), but may also extend to peripheric alternative tissues, including fibroblasts, as previously demonstrated in the case of blood cells [12]. These findings validate fibroblasts as a cell model for the study of sIBM, as previously demonstrated in cases of neurodegenerative and metabolic diseases including AD and T2DM [54,55,56,57].

Moreover, we used these fibroblasts to evaluate mitochondrial contribution into sIBM-T2DM comorbidity by exposing sIBM fibroblasts to different glucose conditions. That way, we observed that control fibroblasts showed full mitochondrial plasticity to adapt to different glucose environments, whilst sIBM fibroblasts were mitochondrially unable to adapt its metabolism. Particularly, sIBM fibroblasts were unable to increase their mitochondrial function in terms of respiration and MRC enzymatic activities in front of changes in glucose bioavailability. Mitochondrial dysfunction in sIBM fibroblasts may lead to reduced bioenergetics and consequent cell compromise. In fact, such metabolic stiffness leads sIBM fibroblasts to increased proton leak and oxidative stress, frequently associated with cell damage [58,59] and disease [9].

To our knowledge, there are no clinical or molecular evidences supporting the preferential affection of muscle tissue in sIBM or the particular targeting of certain muscles (greater in finger flexors and quadriceps), but there is probably a molecular rationale underlying such differential targeting and mitochondria (among others) may stand behind this.

Overall, these data demonstrate that mitochondrial dysfunction limit sIBM cell adaption to changing glucose metabolism, being a source of cell damage and the potential pathophysiological link between sIBM and T2DM. These molecular data confirm clinical comorbidity between both disorders, pointing out sIBM as a new population of underdiagnosed T2DM patients. T2DM is a silent infradiagnosed disease and any clue for emerging groups of risks may account for the reduction of further associated consequences. Interestingly, partial deregulated glucose metabolism was also found in other myositis as PM and DM, thus suggesting common lifestyle cues’ contribution in observed comorbidity.

Probably, both mitochondrial deregulation and lifestyle habits (such as restricted mobility) may be conditioning the manifestation of metabolic disorders such as T2DM in sIBM patients. Due to its idiosyncrasy, myositis are a group of disorders characterized by weakness and wasting of the muscle and consequently low physical activity or a sedentary lifestyle. This situation is associated with the development of several chronic diseases including T2DM. Additionally, due to skeletal muscle contribution to glucose homeostasis and muscle wasting on account of low patient activity, in a kind of vicious circle, glucose metabolism deregulation may be the final fate of a sedentary lifestyle, in this case, triggered by myositis and mitochondrial dysfunction.

Consequently, some general recommendations may be added in the clinical follow-up of patients with myopathy to prevent or manage potential T2DM development in accordance with international standards [60]. They are based on the control of risk factors for T2DM and metabolic syndrome development, including the avoidance of sedentarism and obesity through the performance of exercise (quite restricted in most cases of patients with myositis), but also dietetic control (reduction of caloric intake in cases of obesity, especially carbohydrates).

Some limitations must be acknowledged in the present work. The reduced number of patients included in the study may be the major downfall. Of note, due to the low prevalence of sIBM disorder (considered a rare disease), the study of bigger cohorts is often hampered but recommended. Additionally, further studies are warranted to provide more in-depth understanding of the deregulated pathways underlying glucose impairment in sIBM patients and whether the lipid profile and other features characteristic of the metabolic syndrome are affected in sIBM, even myositis in general, to establish a better correlation between these disorders and the metabolic syndrome. Similarly, the parallel inclusion of fibroblasts from PM or DM patients would have been of interest to deepen the knowledge of common and specific molecular patterns shared among these disorders that clinically overlap.

## 5. Conclusions

In summary, the conclusions of the present work stand for null comorbidity between sIBM and AD but partial comorbidity between sIBM and T2DM through the impairment of glucose homeostasis, mitochondrial function, and probably lifestyle conditions. Although further investigation is needed to validate this pilot study, these findings provide better knowledge of sIBM complications.

On account that effective treatment for sIBM is not yet available, this should be a red flag and the monitorization of the prediabetic state in myositis patients to avoid potential complications in clinical practice associated with the management of both diseases should be considered, thus contributing to improving the quality of life of patients with inflammatory myopathies.

## Figures and Tables

**Figure 1 jcm-09-01446-f001:**
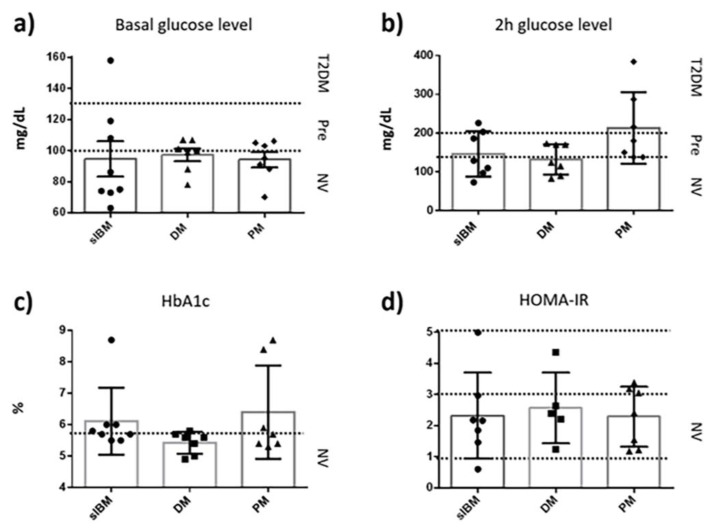
Type 2 Diabetes Mellitus (T2DM) markers analyzed in different myositis populations suggesting frequent alteration in most of the biomarkers when compared to age bracket control data: (**a**) Basal glucose levels; (**b**) Two hours glucose level; (**c**) HbA1c % of glycated haemoglobin; and (**d**) HOMA-IR: Homeostatic model assessment of insulin resistance (normal value 1–3; insulin resistance is consistent with greater than 5 HOMA-IR values). Key: sIBM: sporadic Inclusion Body Myositis; DM: Dermatomyositis; PM: Polymyositis.; Pre: Prediabetic; NV: Normal values.

**Figure 2 jcm-09-01446-f002:**
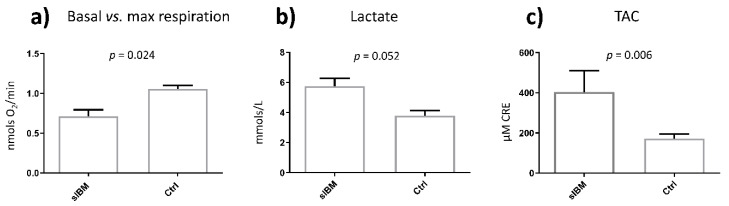
Mitochondrial characterization at standard conditions demonstrated deregulated mitochondrial function in sIBM fibroblasts when compared to healthy controls. (**a**) Oxygen consumption expressed as a ratio of basal respiration vs. max. respiration; (**b**) Lactate secretion expressed in mmols/L and (**c**) TAC (Total Antioxidant Capacity) expressed as µM CRE, Copper Reducing Equivalents. (**b**,**c**) measurements were normalized by total cell amount. sIBM: sporadic inclusion myositis; Ctrl: Healthy controls.

**Figure 3 jcm-09-01446-f003:**
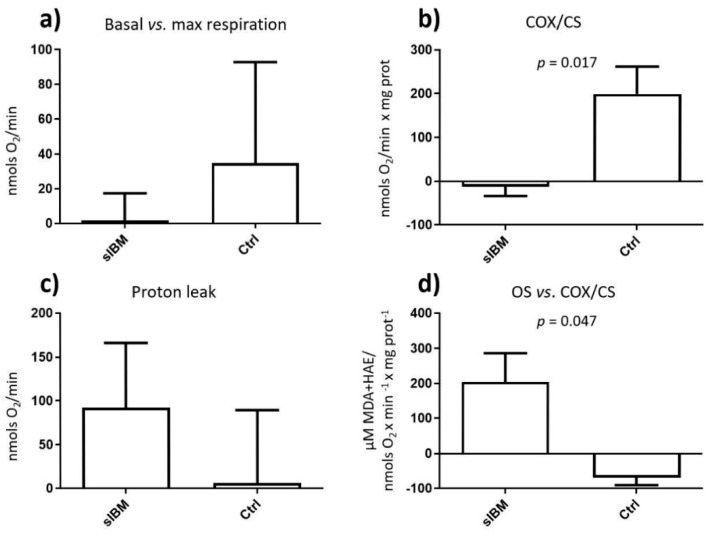
Mitochondrial characterization after glucose changing conditions demonstrating the inability of sIBM fibroblasts to adapt to different glucose conditions when compared to heathy controls. Fold change adaptation from HG (high glucose) to LG (low glucose) for (**a**) Basal respiration vs. max respiration, (**b**) Enzymatic activities (cytochrome c oxidase/citrate synthase; COX/CS), (**c**) Proton leak, (**d**) Oxidative stress (OS) per mitochondrial activity (COX/CS). sIBM: sporadic inclusion myositis; Ctrl: Healthy controls.

**Table 1 jcm-09-01446-t001:** Demographic characteristics of sIBM patients included in the cognitive study.

	Case 1	Case 2	Case 3	Case 4	Case 5	Case 6	Case 7	Case 8	Case 9	Case 10
**Demographics**										
Gender	♂	♀	♂	♂	♀	♂	♂	♀	♀	♂
Age	68.9	72.4	77.4	48.9	67.3	63.0	66.8	81.6	90.5	62.2
Years of education	9	8	18	12	8	11	17	11	7	11
CRQ	11	-	21	12	12	16	18	14	9	21
**Cognitive data**										
Clinical diagnosis	MCI	AD	AD	Control	MCI	Control	MCI	MCI	Dementia	MCI
HADS anxiety/depression	0/4	-	3/3	9/5	6/16	4/1	4/3	9/10	4/5	3/1
MMSE	28/30	19/30	23/30	29/30	26/30	27/30	29/30	24/30	6/30	29/30
Neuropsychological profile	Amnestic (mild)	Global deterioration	Global deterioration	Cognitively normal	Amnestic (mild)	Cognitively normal	Amnestic (mild)	Amnestic (mild)	Severe global deterioration	Amnestic (mild)
**MRI data**										
GCA scale (0–3)	2	1	2	0	1	0	1	-	-	1
MTA score (0–4)	2	3	3	0	1	0	0	-	-	1
Fazekas scale (0–3)	3	1	2	1	1	1	1	-	-	1
**Biological & CSF data**										
*APOE* genotype	ε4/3 ^†^	-	ε3/3	-	ε3/2	ε3/3	-	-	-	ε3/3
Aβ_42_	519.1	-	434.0	-	907.0	634.6	-	-	-	1059.0
Tau	217.9	-	-	-	163.0	136.2	-	-	-	225.0
pTau	40.7	-	182.2	-	39.0	29.6	-	-	-	42.0

Null cognitive impairment was observed in sIBM patients when compared to age bracket control data. Key. CRQ: Cognitive Reserve Questionnaire; HADS: Hospital Anxiety & Depression Scale; MMSE: MiniMental State Examination (20 to 24 score suggests mild dementia, 13 to 20 moderate dementia, and less than 12 indicates severe dementia); GCA: Global Cortical Atrophy; MTA: Medial Temporal lobe Atrophy score (score 0: no atrophy; score 1: only widening of choroid fissure; score 2: also widening of temporal horn of lateral ventricle; score 3: moderate loss of hippocampal volume; score 4: severe volume loss of hippocampus); Fazekas: Fazekas scale for white matter lesions (score 0: none or single punctate white matter hyperintensities; score 1: multiple punctate lesions; score 2: beginning confluency of lesions – bridging; score 3: large confluent lesions); APOE: Apolipoprotein E; CSF: cerebrospinal fluid; Aβ_42_: Amyloid-beta isoform 42; Tau: total tau; pTau: phosphorylated tau. ^†^ APOE ε4 carrier.

**Table 2 jcm-09-01446-t002:** Type 2 Diabetes Mellitus (T2DM) markers in different myositis populations.

	Glu 0 h	Glu 2 h	Ins	Hb1Ac	HOMA-IR	BMI
sIBM1	●					●
sIBM2		●				●
sIBM3						●
sIBM4				●		●
sIBM5	●			●	●	
sIBM6		●		●		
sIBM7	●			●		●
sIBM8		●				●
DM1					●	●
DM2		●				●
DM3	●					●
DM4	●	●				
DM5						●
DM6	●					
DM7		●		●		●
PM1		●		●	●	●
PM2		●				
PM3	●	●		●		
PM4	●	●			●	
PM5		●		●	●	●
PM6	●	●				
PM7		●				

The black spot in one patient for a specific parameter indicates an altered value with respect to bracket control data. These findings suggesting a prediabetogenic state for most of the patients and the deregulation of one or more parameters of the glucose metabolism in all studied myositis. Key: Glu 0 h: basal glucose level; Glu 2 h: venous plasma glucose level at 2 h after the oral glucose tolerance test (OGTT); Ins: basal insuline levels; Hb1Ac: glycated haemoglobin; HOMA-IR: Homeostatic model assessment of insulin resistance; BMI: body mass index, calculated as body weight in kg divided by height in m^2^ and considered altered >25; sIBM: sporadic inclusion myositis; PM: polymyositis; DM: dermatomyositis.

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
