# Peer review of "Mitochondrial Dysfunction: A Common Hallmark Underlying Comorbidity between sIBM and Other Degenerative and Age-Related Diseases"

_jcm, 2020, doi:10.3390/jcm9051446_

Round 1

Reviewer 1 Report

This study aimed to evaluate mitochondrial dysfunction as common clinical and molecular hallmarks among sIBM, AD, and T2DM to find out comorbidity of AD and/or T2DM with sIBM. This is very important and valuable objective for clinical study and the present study was well designed and evidently well written.

However, the small number of patients included is weak point of the present study to conclude the presence or absence of the comorbidity among 3 diseases and if possible, I would like to include more patients for empower the impact of the study.

Suggestions for the minor corrections:

  1. Figure 3 is basic preliminary data for lab and not necessary for the results of the study.
  2. Each 4 graphs in Figure 4 should have identification of the units for Y axis and the figure legends are not matched exactly with figures. The last lines of the figure 4 legenes, from HG: high glucose media~to LG conditions., should be removed or the graphs have HG, LG indications in the Figures.
  3. In the M&M section, line 191-198, authors explained about the control cohort. However, I cannot find out any data or information about age-matched population in the manuscript. Authors should have clinical information of the control group patients as well as the sIBM patients.
  4. I would like to be explained about how the authors excluded the conclusion as like as in between lines 357-359 in page 9. How you normalized with age-paired standard population? And where’s the data?
  5. The same as in the results explained in between lines 361-365. Where are the MRI data of the age-paired subject population.

Author Response

Dear Editor and Reviewers,

On behalf of all the co-authors of the study entitled “Mitochondrial dysfunction: a common hallmark underlying comorbidity between sIBM and other degenerative and age-related diseases” (Manuscript ID: jcm-775798), we would like to express our gratitude to the editor and the reviewers for the careful revision of the manuscript. We sincerely believe that the review has improved the quality of the article, attached to the present cover letter.

As it was considered that the manuscript was not suitable for publication in the previous submitted version, we decided to review the article by including the editor and reviewers' suggestions and resubmitting it to the journal in order to reconsider it for publication in its present form.

The new and revised version has been submitted to the journal by attaching the following files:

  • Main document: Clean version of the revised version of the article, figures and tables.
  • File for review: The same version of the revised article, with all changes highlighted.
  • The present cover letter addressing ‘point by point’ responses to suggested changes made by both the Editor and the Reviewers, with page referenced according to the clean version of the manuscript.

Editor suggested changes:

Please carefully read the guidelines outlined in the 'Instructions for
Authors' on the journal website https://www.mdpi.com/journal/jcm/instructions
and ensure that your manuscript resubmission adheres to these guidelines. In
particular, please ensure that abbreviations have been defined in parentheses
the first time they appear in the abstract, main text, and in figure or table
captions; citations within the text are in the correct format; references at
the end of the text are in the correct format; figures and/or tables are
placed at appropriate positions within the text and are of suitable quality;
tables are prepared in MS Word table format, not as images; and permission
has been obtained and there are no copyright issues.

We have double-checked that the article followed specific journal guidelines and instructions for authors.

Especial attention was made to ensure that abbreviations were defined in parentheses the first time they appear in the abstract, main text, and in figures or tables.

Similarly, in order to meet journal standards, we prepared tables in MS Word format, not as images, initially submitted.

We have additionally checked Language style and grammar questions in order to improve the understanding.

We have additionally corrected the name of one of the co-authors (Pedro J. Moreno-Lozano), that was initially incomplete.

Reviewer 1

Comments and Suggestions for Authors

This study aimed to evaluate mitochondrial dysfunction as common clinical and molecular hallmarks among sIBM, AD, and T2DM to find out comorbidity of AD and/or T2DM with sIBM. This is very important and valuable objective for clinical study and the present study was well designed and evidently well written.

The reviewer has elegantly summarized the main findings and the objective of the study. At this point we would like to thank the effort for the evaluation of the manuscript to the reviewer.

However, the small number of patients included is weak point of the present study to conclude the presence or absence of the comorbidity among 3 diseases and if possible, I would like to include more patients for empower the impact of the study.

We agree with reviewer’s point of view. Reduced sample size is one of the biggest weaknesses of our study. Despite we manage one of the biggest cohorts of sIBM patients as a reference unit in our settings, the main reason for such reduced sample size (n=14) is the low prevalence of sIBM disorder, considered a rare disease (ORPHA611). Actually, in the present study we have included all patients willing to participate from our hospital. Consequently, further inclusions could only be considered in case on new diagnosis or multicentric studies, not approachable within the 7-10 day period of review, particularly with the present pandemic restrictions. We have reflected such limitation in lines 509-511 from the manuscript as follows: ‘The reduced number of patients included in the study may be the major downward. Of note, due to the low prevalence of sIBM disorder (considered a rare disease), the study of bigger cohorts is often hampered but recommended’.

Suggestions for the minor corrections:

1. Figure 3 is basic preliminary data for lab and not necessary for the results of the study.

 According to reviewer’s suggestion we have moved Figure 3, corresponding to preliminary data, to supplementary material, as Supplementary Figure 1 (see page 14) and we have consequently re-numbered the rest of figures in the new version of the manuscript.

2. Each 4 graphs in Figure 4 should have identification of the units for Y axis and the figure legends are not matched exactly with figures. The last lines of the figure 4 legends, from HG: high glucose media~to LG conditions., should be removed or the graphs have HG, LG indications in the Figures.

Thanks to reviewer’s suggestions we have addressed all these changes in Figure 4 panels and legends to improve the understanding and avoid redundant information. See pages 14-15.

3. In the M&M section, line 191-198, authors explained about the control cohort. However, I cannot find out any data or information about age-matched population in the manuscript. Authors should have clinical information of the control group patients as well as the sIBM patients.

We sincerely acknowledge reviewer’s advice. In the current version of the article we clarify how we normalized data. It was wrongly expressed. We don’t have a control group parallelly included to sIBM patients for T2DM and AD diagnosis. We compare results of sIBM patients with control data stratified by age (age bracket control data). This is the standard procedure for T2DM and AD diagnosis, and this is how we evaluated sIBM patients in the present study, with the assistance of trained staff from our hospital dedicated to that aim. However, to evaluate potential overlap with other myositis, we evaluated PM and DM patients for T2DM and AD (in parallel to sIBM), to compare obtained results to sIBM patients. Additionally, experimental studies of mitochondrial function required parallel inclusion of patients (sIBM) and, in this case, healthy controls (C). We have clarified data normalization in M&M section (see lines 194-203 page 5) and provided the main epidemiologic characteristics of PM, DM and healthy control cohorts in the Results section (lines 347-350 page 8).

4. I would like to be explained about how the authors excluded the conclusion as like as in between lines 357-359 in page 9. How you normalized with age-paired standard population? And where’s the data?

Again, thanks to reviewer’s comments, we have clarified normalization of data from the study. Results of clinical and neuropsychological assessment were compared to standards, as those from Bacigalupo et al., 2018, corresponding to age bracket control data (see new reference 45). We didn’t include prospectively a control group of healthy controls for AD evaluation, as explained in page 5 lines 194-197 from the M&M section. This is the standard protocol for AD diagnosis.

In our study, 3 subjects presented symptoms of anxiety and/or depression. MMSE ranged between 6 and 29 (maximum MMSE score in health is 30 points) and the neuropsychological profile of the study subjects was heterogeneous, although most of them showed an amnestic pattern (verbal and/or visual memory decline). Eight out of the ten subjects that underwent clinical and neuropsychological evaluation were cognitively impaired.  From those, 5 were diagnosed with MCI and 3 with dementia (two of them were diagnosed as AD). See clinical and neuropsychological data of the Neurological study in Table 1.

However, and despite reported disturbances in the clinical and neuropsychological assessment, no higher prevalence of dementia or AD was detected in sIBM patients when results were compared to prevalence of dementia in Europe (Bacigalupo et al., 2018). 

Reported findings have re-written in the Results section and corresponding bibliography added to better express the output of the study (see lines 366-369 page 10).

5. The same as in the results explained in between lines 361-365. Where are the MRI data of the age-paired subject population.

Again, thanks to reviewer’s comments, we have clarified normalization of data from the study. Image findings were compared to standards corresponding to age bracket control data; we didn’t include prospectively a control group of healthy controls for AD evaluation, as previously explained and now corrected in page 5 lines 194-197. This is the standard protocol for AD diagnosis.

MRI analysis was performed in 8 sIBM patients (see Table 1). Medial temporal lobe atrophy (MTL score > 1) was found in 3 subjects (maximum pathological score is 4 points) and all individuals presented different levels of white matter deterioration, including two subjects with a Fazekas score 2 (maximum pathological score is 3). However, MTL score > 1 was also found in controls in previous studies (Pereira et al., 2014; see new reference 46). Furthermore, in our study 1 out of 2 subjects with MTL >1 had a normal AD CSF biomarkers, so excluding AD (ref 27). Thus, we concluded that MRI analysis of sIBM patients showed no higher incidence of AD. Reported findings have re-written and corresponding bibliography added to better express the findings of the study (see lines 375-378 page 10).

Reviewer 2 Report

The authors investigated neurological and metabolic status in patients with sIBM to determine the comorbidity of sIBM with AD or T2DM, and evaluated mitochondrial homeostasis in sIBM fibroblasts. They found null comorbidity between sIBM and AD but partial comorbidity between sIBM and T2DM through the impairment of glucose homeostasis, mitochondrial function and probably lifestyle conditions. They concluded that monitorization of the prediabetic state to avoid potential complications in the clinical practice may contribute to improve the quality of life of patients with inflammatory myopathies. This is an interesting and important study, but there are several concerns with this paper.

1) The authors found 5 patients diagnosed with MCI, and 3 with dementia (including 2 with AD) in their sIBM cohort. However, they mentioned no higher incidence of AD was detected in sIBM patients after normalization with age-paired standard population. Statistical comparison should be shown between the sIBM cohort and age-matched population. How many patients had been concomitantly diagnosed as sIBM and AD before their investigation.

2) Similarly, in T2DM evaluation, the frequencies of abnormal patients in each work-up should be statistically compared between the sIBM cohort and age-matched population.

3) The authors should explain the derivation of control fibroblasts in mitochondrial respiration study: a healthy volunteer or a disease-control patient? To figure out the involvement of mitochondrial dysfunction, fibroblasts from DM and/or PM should be examined in the mitochondrial respiration study or other ones.

4) The authors found sIBM fibroblasts were mitochondrially unable to adapt its metabolism. Are there any evidences suggesting the abnormal function of fibroblast in the pathogenesis of sIBM? Finger flexor and quadriceps muscles are preferentially affected in sIBM whereas other muscles are comparatively preserved until the advanced stage. Can the preferential muscle involvement be explained by mitochondrial dysfunction?

5) For monitorization and treatments of the prediabetic state in patients with sIBM, a proposal for efficient work-up and medications would be helpful for clinicians associated with the clinical practice.

Author Response

Dear Editor and Reviewers,

On behalf of all the co-authors of the study entitled “Mitochondrial dysfunction: a common hallmark underlying comorbidity between sIBM and other degenerative and age-related diseases” (Manuscript ID: jcm-775798), we would like to express our gratitude to the editor and the reviewers for the careful revision of the manuscript. We sincerely believe that the review has improved the quality of the article, attached to the present cover letter.

As it was considered that the manuscript was not suitable for publication in the previous submitted version, we decided to review the article by including the editor and reviewers' suggestions and resubmitting it to the journal in order to reconsider it for publication in its present form.

The new and revised version has been submitted to the journal by attaching the following files:

  • Main document: Clean version of the revised version of the article, figures and tables.
  • File for review: The same version of the revised article, with all changes highlighted.
  • The present cover letter addressing ‘point by point’ responses to suggested changes made by both the Editor and the Reviewers, with page referenced according to the clean version of the manuscript.

Editor suggested changes:

Please carefully read the guidelines outlined in the 'Instructions for
Authors' on the journal website https://www.mdpi.com/journal/jcm/instructions
and ensure that your manuscript resubmission adheres to these guidelines. In
particular, please ensure that abbreviations have been defined in parentheses
the first time they appear in the abstract, main text, and in figure or table
captions; citations within the text are in the correct format; references at
the end of the text are in the correct format; figures and/or tables are
placed at appropriate positions within the text and are of suitable quality;
tables are prepared in MS Word table format, not as images; and permission
has been obtained and there are no copyright issues.

We have double-checked that the article followed specific journal guidelines and instructions for authors.

Especial attention was made to ensure that abbreviations were defined in parentheses the first time they appear in the abstract, main text, and in figures or tables.

Similarly, in order to meet journal standards, we prepared tables in MS Word format, not as images, initially submitted.

We have additionally checked Language style and grammar questions in order to improve the understanding.

We have additionally corrected the name of one of the co-authors (Pedro J. Moreno-Lozano), that was initially incomplete.

Reviewer 2

Comments and Suggestions for Authors

The authors investigated neurological and metabolic status in patients with sIBM to determine the comorbidity of sIBM with AD or T2DM, and evaluated mitochondrial homeostasis in sIBM fibroblasts. They found null comorbidity between sIBM and AD but partial comorbidity between sIBM and T2DM through the impairment of glucose homeostasis, mitochondrial function and probably lifestyle conditions. They concluded that monitorization of the prediabetic state to avoid potential complications in the clinical practice may contribute to improve the quality of life of patients with inflammatory myopathies. This is an interesting and important study, but there are several concerns with this paper.

The reviewer has smartly summarized the aims and main findings of the study. We would like to thank the evaluation of the manuscript to the reviewer that, in our opinion, improved the understanding and interpretation of data.

1) The authors found 5 patients diagnosed with MCI, and 3 with dementia (including 2 with AD) in their sIBM cohort. However, they mentioned no higher incidence of AD was detected in sIBM patients after normalization with age-paired standard population. Statistical comparison should be shown between the sIBM cohort and age-matched population. How many patients had been concomitantly diagnosed as sIBM and AD before their investigation.

We sincerely acknowledge reviewer’s advice that avoided a potential pitfall. We must acknowledge that we wrongly expressed the normalization of data. We don’t have a control group of healthy subjects parallelly included to sIBM patients for T2DM and AD diagnosis. As we didn’t perform a case-control study, we couldn’t perform a statistical comparison of data. We compare results of sIBM patients with age bracket control data. In case of AD, reference data was extracted from Bacigalupo et al., 2018, Pereira et al., 2014 and Albert et al 2013 (see new references 45, 46 and previous reference 27). This is the standard procedure for AD (and T2DM) diagnosis, and this is how we evaluated sIBM patients in the present study, with the assistance of trained staff from our hospital dedicated to that aim. We have addressed M&M and Results sections where we expressed the normalization of T2DM and AD results (see lines 194-197 page 5).

Interestingly, before our investigation, none of these patients had been diagnosed of T2DM or AD, as pinpointed by the reviewer. This statement has been added in the Discussion section of the paper (see page 15 line 449). Our study demonstrated that no higher incidence of AD was detected in sIBM patients considering globally our data (clinical prevalence of dementia, MRI data and typical AD CSF biomarkers). For instance, we found only 1 out 5 patients had a typical AD CSF biomarkers. This percentage is very similar to the frequencies of brain amyloidosis found in individuals with normal cognitive function at similar ages (Jack et al., 2014, see new reference 47). Reported findings have re-written and corresponding bibliography updated to better express such findings (see lines 386-391 pages 10-11).

2) Similarly, in T2DM evaluation, the frequencies of abnormal patients in each work-up should be statistically compared between the sIBM cohort and age-matched population.

We must acknowledge again the same caveat when defining how we normalized data, this time in regards to T2DM. Thanks to the reviewer, now the procedure (standard in diagnosis) is corrected and detailed in the M&M section from the new version of the article (see lines 194-197 page 5). Unfortunately, this approach for clinical evaluation of T2DM is incompatible with statistical analysis with respect to case-control studies.

3) The authors should explain the derivation of control fibroblasts in mitochondrial respiration study: a healthy volunteer or a disease-control patient? To figure out the involvement of mitochondrial dysfunction, fibroblasts from DM and/or PM should be examined in the mitochondrial respiration study or other ones.

Again, please apologize the poor definition of normality ranges used to compare patient outputs. Thanks to reviewer’s advice, now the proper definition for control fibroblast derivation is included in the new version of the M&M section from the manuscript (see lines 201-203 page 5), corresponding to healthy volunteers donation.

We must agree that the parallel inclusion of PM and DM fibroblasts to evaluate mitochondrial function compared to sIBM cell phenotyping would had definitively been a strength of the present study and, thus, we have admitted its value in the Discussion section of the novel version of the manuscript, as follows (see lines 516-518 page 16): ‘Similarly, the parallel inclusion of fibroblasts from PM or DM patients would had been warranted to deepen in the common and specific molecular patterns shared among these disorders that clinically overlap’. Unfortunately, we wouldn’t be able to perform such evaluation in the 7-10 day period of review, additionally, with all the restricted access to laboratory facilities on account of the COVID19 pandemia.

4) The authors found sIBM fibroblasts were mitochondrially unable to adapt its metabolism. Are there any evidences suggesting the abnormal function of fibroblast in the pathogenesis of sIBM? Finger flexor and quadriceps muscles are preferentially affected in sIBM whereas other muscles are comparatively preserved until the advanced stage. Can the preferential muscle involvement be explained by mitochondrial dysfunction?

Previous to the present study, there were no evidences of the abnormal function of fibroblasts (any in general of peripheral tissues) in sIBM patients (apart from the target tissue, muscle). This is one of the main findings of this work (the validation of this cell model), in addition to trends towards T2DM-sIBM comorbidity. The study of sIBM fibroblasts unveil not only the mitochondrial implication in disease but also the systemic affection of this myositis, always thought to be restricted to the muscle (see Discussion section, lines 468-473 page 15).

Both in dystrophies and myopathies there is selective target of specific muscles. This is the case of finger flexors and quadriceps muscles for sIBM, but also face and shoulder girdle affection in case of facioscapulohumeral dystrophy or pharynx and palpebral muscle affection in case of oculopharyngeal dystrophy. Up to our knowledge, there are no evidences supporting such specific targeting, but probably, there is a molecular reason for that, and mitochondria (among others) may be underlying such differential targeting. Such statement has been added to the Discussion section (see lines 483-486 page 16), thanks to reviewer’s advice, thus enriching the output of the present findings, as follows: ‘Up to our knowledge, there are no clinical or molecular evidences supporting the preferential affection of muscle tissue in sIBM, or the particular targeting of certain muscles (greater in finger flexors and quadriceps), but there is probably a molecular rational underlying such differential targeting, and mitochondria (among others) may stand behind’.

5) For monitorization and treatments of the prediabetic state in patients with sIBM, a proposal for efficient work-up and medications would be helpful for clinicians associated with the clinical practice.

We definitively agree that the addition of clinical recommendations to prevent or manage T2DM in patients with myositis would be of help for clinicians, thus, we have included general recommendations in accordance to International standards. They are mainly based in the control of risk factors for T2DM and metabolic syndrome development including the avoidance of sedentarism and obesity through the performance of exercise (quite restricted in case of most of patients with myositis), but also dietetic control (extracted from Standards of the American diabetes association, published in Diabetes Care 2020; see new reference 60). These recommendations have been included in the new version of the Discussion section from the manuscript (see page 16 lines 503-508). We would like to thank the reviewer for such suggestion.

Round 2

Reviewer 2 Report

The points raised by the reviewer have been addressed sufficiently. I would like to congratulate the authors on their interesting study.